# SEATANI: hazards from seamounts in SouthEast Asia, Taiwan, and Andaman and Nicobar Islands (eastern India)

Andrea Verolino[1], Su Fen Wee[2], Susanna F. Jenkins[1,2], Fidel Costa[1,2,3], Adam D. Switzer[1,2]

[1]Earth Observatory of Singapore, Nanyang Technological University, 50 Nanyang Ave, Singapore, 639798, Singapore

[2]Asian School of Environment, Nanyang Technological University, 50 Nanyang Ave, Singapore, 639798, Singapore

[3]Institut de Physique du Globe de Paris, Université Paris Cite, CNRS, 1 Rue Jussieu, Paris, 75005, France

Correspondence to: Andrea Verolino
Email: andrea.verolino@ntu.edu.sg

**Abstract.** Submarine volcanism makes up approximately 85% of volcanism taking place on Earth, and its eruptions have the potential to cause several hazards, including ash dispersal, pumice rafts, pyroclastic density currents, sector collapses and tsunamis. Recent examples include the eruptions in Japan and in the Kingdom of Tonga in 2021 and 2022 respectively, but there has been little to no study of submarine volcanism in Southeast Asia and its surroundings. Here we provide a compilation of 466 seamounts from the region, from different published sources, through the SEATANI dataset (Southeast Asia + Taiwan + Andaman & Nicobar Islands). We use this newly compiled dataset to assess on a regional level the seamount hazard potential and exposure potential as a springboard for future more quantitative hazard studies for the region. The hazard potential was assessed through seamount morphological/structural analyses, to determine the seamount evolution stage and, grade of maturity. The exposure potential was evaluated with two different approaches: An areal analysis of the number of assets within a 100 km radius of each seamount; and the development of a hazard-weighted seamount density map to highlight potential areas of interest for future more-in-depth studies. Our results show that there are several potentially hazardous seamounts in this region, and Taiwan had the highest hazard and exposure potential, for all assets considered, while Philippines, Indonesia and Vietnam have relatively high exposure potential for submarine communication cables and ship traffic density. The results from this work serve as a first step for Southeast Asian and neighbouring countries to become more resilient against and prepared for submarine volcanic eruptions in the region.

## 1 Introduction

Volcanic seamounts are submerged or mostly submerged volcanoes, and can be defined as "*any geographically isolated topographic feature on the seafloor taller than 100 m, including ones whose summit regions may temporarily emerge above sea level*" (Staudigel et al., 2010). The number of volcanic seamounts around the world is in the order of tens of thousands. A recent estimate suggests that there are ~35,000 seamounts >400-m in height (Gevorgian et al., 2023); however, limitations in detection suggest that this is a significant underestimate, especially in shallow continental shelf regions close to land masses (Kim and Wessel, 2011). Volcanic seamounts are generally detected through satellite-derived altimetry and gravimetry, however, these methods are limited by

resolution (i.e. kilometric scale, not allowing for the detection of small seamounts), and noise in the gravimetry
measurements in areas with thick sequences of sediments (e.g. within continental margins; Kim and Wessel,
2011). It is likely that there are many more seamounts globally than those we are aware of.
Volcanic seamounts, particularly deep-sea ones, have been traditionally considered to be a negligible threat to
society (Cas, 1992; Whelley et al., 2015), for several reasons. The first is that most of them are completely
underwater, hence they are economically and logistically difficult to monitor, map and sample compared to their
subaerial counterparts; as a result, their eruption frequency and intensity have not been properly assessed. A
second reason is that they are often located far from major landmasses, hence not considered an imminent threat
to populations. Thirdly, most of them have their summit in deep waters (> 3000 m b.s.l.), and this makes them
hypothetically less hazardous than shallower volcanoes, because of the high hydrostatic pressure hindering
explosivity. Finally, most ocean intraplate volcanoes, particularly those approaching subduction zones, are likely
to be extinct and have not been active for millions of years (Staudigel and Clague, 2010), hence, have been not
considered of interest for volcanic hazard. As a result, seamounts are vastly understudied around the world. In
January 2022, the eruption of Hunga volcano in the SW Pacific (Kingdom of Tonga), demonstrated that erupting
seamounts can have a large impact on people and their activities, even in a remote location such as the southwest
Pacific Ocean. The eruption produced the highest volcanic plume ever recorded (~58 km) (Taha et al., 2022),
unusually fast tsunamis that travelled across the Pacific Ocean for thousands of kilometres (Gusman et al., 2022),
and damage of millions of USD across the entire region, with the Kingdom of Tonga being the most affected
(damage equivalent to ~19% of the national GDP, and several casualties recorded) (The World Bank, 2022).
A few notable seamounts around the world have been studied in detail using multibeam surveys to obtain high
resolution bathymetry (up to 1-m), and in a few cases were accompanied by sampling and/or video recording of
the eruptions from Remotely Operated Vehicles (ROV's). Typically, these investigations occur after an impactful
eruption. Examples include Havre volcano, (Kermadec arc), NW Rota-1 (Mariana arc), West Mata (Tonga arc),
Fani Maoré (NW of Madagascar), Axial caldera (Juan de Fuca ridge) (Carey et al., 2014; Murch et al., 2019a;
Dürig et al., 2020; Embley et al., 2006; Chadwick et al., 2008; Schnur et al., 2017; Clague et al., 2011; Dziak et
al., 2015; Murch et al., 2022; Feuillet et al., 2021; Hammond, 1990; Caress et al., 2012; Clague et al., 2013),
among others. All the above-mentioned submarine volcanoes were surveyed as part of large, well-funded,
multidisciplinary projects that provided a wealth of data (bathymetry, rock geochemistry, tephra granulometry
and componentry, etc.). It is logistically impossible to apply the same approaches to the thousands of seamounts
worldwide, therefore, a regional approach is needed to characterise seamounts in a simple and efficient manner
that allows for a broad focus on lesser-known areas potentially at risk.
Past global studies on volcanic seamounts have included classifications based on morphology or growth stage
of the edifice (Schmidt et al., 2000; Wessel, 2007; Staudigel and Clague, 2010; Kim and Wessel, 2011; Gevorgian
et al., 2023). Some authors found direct relationships between seamount morphometric parameters (e.g. basal
width and height) and linked them to the tectonic setting (Schmidt et al., 2000; Gevorgian et al., 2023). These
classifications, however, have never been used for assessing hazard potential or exposure on a regional scale.
For the region of Southeast Asia (SEA), there has been some effort in assessing hazards from what we define
here as volcanic seamounts; examples include Krakatau and Banua Wuhu, Indonesia, and Didicas, Philippines
(Hamzah et al., 2000; Paris et al., 2014; Mutaqin et al., 2019; Hidayat et al., 2020; Zorn et al., 2022; NCEI/WDS,
n.d.), however, these studies focused on volcanic islands, and there is little or no consideration for the hazard
potential from fully submerged volcanoes.
Our newly compiled dataset includes 466 seamounts from different sources (Fig. 1, Table S1 – Supplementary
information) and enclosed within three (Indonesia, Philippines and Vietnam) of the nine Exclusive Economic
Zones (EEZs) of Southeast Asia (Brunei, Burma, Cambodia, Indonesia, Malaysia , Philippines, Singapore,
Thailand and Vietnam) and the neighbouring countries Taiwan and eastern India (Andaman and Nicobar Islands),
and has been named SEATANI (**SEA** + **T**aiwan + **A**ndaman & **N**icobar **I**slands). This dataset includes both fully
submerged volcanoes and some small volcanic islands, whose submerged portion makes up most of the edifice.
This region is interesting for several reasons: (i) It is very volcanologically active, but little is known about its
underwater features; (ii) Millions of people live along its coasts; (iii) There are infrastructure worth billions of
dollars on the seafloor of the target area (e.g. submarine telecommunication cables) (Wang et al., 2019); and (iv)
It has a rather high density of ship traffic. In this paper, we have two main goals: one is to characterise the
seamount morphology and evolution stage and link them to the *hazard potential* for seamounts in the region,  and
the other is to highlight areas of high *exposure potential*, to motivate and focus future studies. To accomplish the
first goal of characterising seamounts and assess their hazard potential, we conduct qualitative (seamount type:
caldera, guyot, simple cone, composite cone) and quantitative morphological analysis (height, summit water
depth) by using open-access bathymetry datasets (e.g. Gebco 2021; NOAA DEM Global Mosaic; NOAA
Multibeam Bathymetry Mosaic). Additionally, we also conducted a more qualitative analysis based on higher
resolution bathymetry (Multibeam data from NOAA – 90m/pixel), where we highlight key seamount features
(e.g. submarine landslides, explosive craters, new seamounts) that otherwise would not be detected from Gebco
or the NOAA DEM Global mosaic dataset. Despite the multibeam data having limited regional coverage (< 10%),
they reveal significant seafloor morphologies that can motivate future quantitative hazard assessments for the
region, e.g. numerical hazard modelling. The second goal of highlighting areas of high exposure potential is
achieved through two types of analysis, a quantitative one, where the number of assets and activities (population,
submarine fibre-optic cables, and ship traffic density) within 100 km of volcanic seamounts is counted; and a
semi-quantitative one, where the hazard potential of each seamount is used to weight the potential areal hazard
extent for the entire region of interest. We acknowledge that our work has some limitations, in particular, the
hazard and exposure potential are not quantified based on geological, geochemical, tectonic setting, age, and
frequency/magnitude information, which are indeed needed for more quantitative studies (a focused discussion is
provided later in the text to address these points). However, our intent with this work is to provide the basic but
fundamental elements for future more quantitative studies.

## 2 Methods

### 2.1 Compilation of SEATANI

Following the seamount definition from Staudigel et al. (2010), here only used for volcanic seamounts, we pre-
compiled a list of seamounts for the region of interest by using three different types of sources: 1) The GVP
database (Global Volcanism Program 2013), where we include seamounts that have erupted from the Pleistocene
(n= 42); 2) The seamount dataset Gevorgian et al. (2023) (n= 405), which is an updated version of the dataset
from Kim and Wessel (2011), where they used statistical methods to differentiate volcanic from non-volcanic
seamounts; and 3) Seamounts from individual studies around the southeast Asian region found in literature (n=

35) (Li et al., 2013; Fan et al., 2017), which have been detected through geophysical methods (i.e. interpretation of seismic profiles), for a total of 482 entries for the region considered. The definition proposed by Staudigel and colleagues, however, does not provide specific directions on islands (at what extent a volcanic island is still considered a seamount). Therefore, in order to guarantee reproducibility and to maintain our broad focus on the unknown hazard potential of seamounts, we did not include islands whose emerged volume was > 30% of the total seamount volume, and/or their maximum elevation was > 1000 m above sea level (a.s.l.) (more details on this methodology are provided in the supplementary information). Following this criterion, none of the seamounts from the Gevorgian et al. (2023) dataset or from the literature studies were removed, however, 16 GVP volcanoes were excluded (Table S2, Supplementary information), bringing the total to 466 volcanic seamounts. Despite the choice of 30% and 1000 m a.s.l. was somewhat arbitrary, it allows comparisons across studies and is in line with our focus here, which is primarily on submarine volcanoes.

**2.2 Bathymetry and exposure datasets**

For the bathymetry, we used different datasets of different resolution based on each specific purpose. These include *Gebco 2021*, *DEM global mosaic* (from NOAA/NCEI), and *Multibeam Bathymetry Mosaic* (from NOAA/NCEI). Gebco 2021 is a gridded bathymetric dataset with interval grid of 15 arc-second (450-m/pixel), and was used for the quantitative morphological classification (seamount growth stages) and exposure potential analyses (quantitative and semi-quantitative). Despite the relatively low/medium resolution, it has global coverage with bathymetry data deriving from different acquisition methods (Fig. S1, supplementary material), and was clipped for the region of interest (North: 36.5°, South: -14.3°, West: 82.0°, East: 145.6°). The DEM global mosaic is a colour shaded relief raster file that was exclusively used for the qualitative morphological classification (seamount morphotypes); it is a seamless bathymetry/topography mosaic that combines DEMs from several sources (e.g. direct and indirect measurements from ships and satellites) and different resolutions (450-m/pixel or better), with the higher-resolution DEMs displayed on top of the lower resolution ones (where both available). Since DEMs of different resolution cannot be extrapolated from this file, but must be downloaded individually, we used the mosaic format for efficiency and for visualization purposes only. The file was clipped with the same extent as Gebco 2021, for consistency. The Multibeam Bathymetry Mosaic is the dataset with the highest resolution among the datasets used here (90-m/pixel); it is a gridded colour shaded relief, deriving from multibeam survey data collected over the years (from ~1980 to present). This dataset has a coverage lower than 10% for the region of interest, therefore was only used for qualitative image analyses, both for the morphological classification (in combination with the DEM Global mosaic) and for the characterization of bathymetric features (see discussion) at some of the locations enclosed within our study area (where there was data coverage).

To assess the exposure potential, we used different open-access datasets for population, submarine communication cables and ship traffic density. We chose these assets for three reasons: (i) Data were available for quantitative analyses on GIS environment on a regional scale; (ii) We considered them as the assets potentially more exposed to multiple hazards from a submarine volcanic eruption in a regional perspective (e.g. air traffic exposure was not quantified here because potentially only exposed to development of a tephra column); And (iii) such exposure assessment from submarine volcanic eruptions has not been done before, particularly for submarine communication cables and ship traffic density, which instead have shown to be vulnerable elements to natural hazards (Ohno et al., 2022; Speidel, 2022). For population estimates, we used LandScan (Sims et al., 2022), which

has a spatial resolution of around 1 km (30 arc-seconds) and has been widely used in previous volcanic hazard
assessments (Reyes-Hardy et al., 2021; Jenkins et al., 2022; Verolino et al., 2022a). For the submarine cables we
used data from *TeleGeography* (last update in 2017); while for the ship traffic density we utilised data from *The*
*World Bank Group*, which reports hourly Automatic Identification Systems (AIS) positions, recorded between
January 2015 and February 2021, at a spatial resolution of 500-m/pixel. This dataset included separate files for
commercial, leisure, passenger, oil and gas, and fishing vessels respectively, however, we used the combined file,
assuming no distinction across vessel types.

**2.3 Volcanic seamounts classification**
Volcanic seamounts were classified through two different approaches: 1) Qualitative, based on seamount shape
(i.e. caldera, guyot, simple cone, composite edifice; Fig. 2 and Table 2); and 2) Quantitative, based on the
seamount height and depth that gives a stage of growth, as proposed by Staudigel and Clague (2010) (Stage 1 to
5: defined in Table 3). Both classifications were obtained from analyses conducted in GIS environment (Esri®
ArcMap 10.7.1). For the qualitative classification, we overlaid the seamount locations over the bathymetry
datasets, and conducted visual image analysis to establish morphotype (Table 2, Fig. 2), using the highest
resolution available for that area (NOAA DEM Global Mosaic, 450-m resolution or better, or NOAA Multibeam
data, 90-m resolution). This morphological assessment was conducted by authors A. Verolino and S.F. Wee, to
test for consistency and reproducibility, as the classification can be partially subjective. For the quantitative
method, we applied the Staudigel and Clague (2010) classification (used here for the first time for hazard
purposes) and obtained the seamount maximum summit height and base water depth within a 30 x 30 km bounding
box of the given seamount location (following Kim and Wessel, 2011). These were in turn used to assign a stage
of growth. Staudigel and Clague (2010) also included Stage 6 seamounts (those approaching the trench of a
subduction zone, or that already started being subducted), however, to maintain the growth stage classification in
a state that is as quantitative as possible, we included them within the low hazard potential i.e. Stage 1, 2 or 5
seamounts (i.e. deep-water or extinct seamounts). We did this depending on their height and water depth though
we know that the Staudigel and Clague (2010) classification, is not specific on how close to the subduction trench
a seamount must be to be considered stage 6, leaving some subjectivity in the classification). We used both
qualitative and quantitative classification approaches in parallel to obtain different types of information
(morphological and growth stage); however, for exposure calculation we refer only to the quantitative approach
(i.e. growth stage).

**2.4 Exposure analysis**
We conducted two types of exposure potential assessments: 1) A quantitative analysis of population, submarine
communication cables and ship traffic density within 100 km from each seamount; and 2) A semi-quantitative
assessment, through a hazard-weighted seamount density map, to assess what countries are more likely to be
threatened by a seamount within the study region.

194        For the first type of assessment, we chose concentric 100 km radii to include exposure potential of the above-

mentioned assets with the approximation that this would include the more damaging processes from most volcanic
hazards (e.g. tephra fallout, PDCs, sector collapses). This choice is in line with previous regional volcanic threat
studies (Small and Naumann, 2001; Brown et al., 2015), however, we acknowledge that using concentric radii is
an oversimplification of volcanic hazard extents (Jenkins et al., 2022).

The semi-quantitative assessment considered the concentration of seamounts, weighted by their hazard rank (Table 1), and highlights regions of higher hazard potential. We created a weighted seamount density map (Kernel Density Estimation, KDE), based on the seamount stage of growth, with the assumption that more heavily weighted seamounts have a greater hazard potential. The KDE was performed on Esri® ArcMap 10.7.1, which assigns a default bandwidth in function of the input dataset (~630 km in this case), and proven to be reliable in previous exposure studies (Verolino et al., 2022a). To verify whether the default bandwidth was suitable to our case, we conducted additional KDEs by manually assigning different bandwidths (named "search radius" in ArcMap) in the range 800-100 km respectively; the results, reported in the supplementary material (Fig. S2), support our choice to use the default bandwidth. The choice of weight assigned to each growth stage (Table 1) was based on the Global Historical Tsunami Database (NCEI/WDS), where out of 164 historical volcanic tsunamis (from 1610 BC to present), 115 were from volcanic seamounts; of these, 78% (n= 90) were from stage 4, 20% (n= 23) from stage 3, and nearly 2% (n= 2) from stage 1, 2, 3 or 5 seamounts (depth of seamount unknown). To compensate the paucity of historical information/data from stages 3, 2, 1 and 5 (shallow or deep), compared to stage 4 seamounts (partially emerged), and to include volcanic hazards as well, we arbitrarily adjusted these percentages to 60% and 35% for stage 4 and stage 3 respectively, and the rest was distributed through stage 1, 2 and 5 seamounts (Table 1). Exposure potential was then assessed based on the extent of high-density area (higher exposure potential: $> 2.9 \times 10^{-6}$ seamounts/km$^2$) obtained from the KDE.

**Table 1. Seamount hazard ranking based on the Global Historical Tsunami Database (NCEI/WDS) and growth stage from Staudigel and Clague (2010).**

| Seamount hazard ranking | Seamount growth stage | Historical volcanic tsunami occurrence (%) | Hazard weight |
|---|---|---|---|
| *1* | 4 | 78 | 0.6 |
| *2* | 3 | 20 | 0.35 |
| *3* | 2 | | 0.03 |
| *4* | 1 | 2 | 0.01 |
| *5* | 5 | | 0.01 |

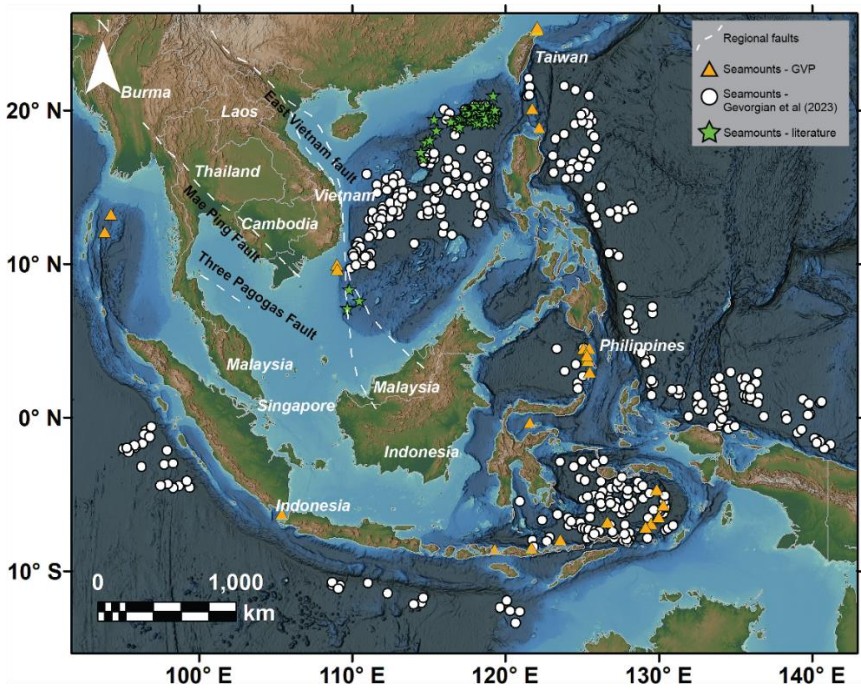

**Figure 1. Map of the study area with seamounts locations and major regional faults.
Basemap is from NOAA (DEM Global Mosaic).**

## 3 Linking seamounts morphology and evolution stage with their potential hazards

### 3.1 Seamount morphotype

Seamount morphology can be used to infer information about the seamount eruptive history. An important consideration is that once seamounts are completely submerged, they do not experience major erosion, retaining most of their original morphological constructive features, unless new eruptions and/or disruptive events such as landslides take place. Therefore, classifying seamounts based on their large-scale morphological features overcomes the resolution issue that we generally have at smaller scales. Below in Table 2 (examples shown in Fig. 2), we provide general guidelines we used for the classification of seamount morphotypes. A background for each morphotype, with a link to their hazard potential and with relevant examples from the literature, is provided in the supplementary information file.

Table 2. Seamount morphotype descriptions.

| Seamount morphotype | Description |
|---|---|
| Simple Cone | Regular-shaped and conical pointy volcanic edifice with only one vent |
| Composite edifice | Irregularly shaped volcanic edifice with one or more vents. This morphotype also includes ridges and flank ephemeral cones (e.g. subaqueous portion of a volcanic island) |
| Caldera | Volcanic edifice with prominent central depression with diameter ~ 4-8 km |
| Guyot | Flat-topped volcanic edifice with relatively steep flanks |

### 3.2 Seamount growth stage

Staudigel and Clague (2010) classified seamounts based on their growth stage (Stage 1 to 5), here we use the same approach to first assign a growth stage to the SEATANI seamounts, and then link the growth stage to a given potential hazard(s) that may be common for that particular growth stage. In Table 3 we report the main

characters for each growth stage (from Staudigel and Clague, 2010), and associated potential hazards (Murch
et al., 2019a; Paris et al., 2014; Clague et al., 1990; Harders et al., 2014; Verolino et al., 2018, 2019, 2022b;
Jutzeler et al., 2014; Deardorff et al., 2011; Omira et al., 2016; Newland et al., 2022). A more comprehensive
analysis of seamount growth stages and their potential hazards is provided in the supplementary information file.

**Table 3. Seamount growth stage and associated potential hazards. Adapted from Staudigel and Clague (2010).**

| Seamount growth stage | Description (from Staudigel and Clague, 2010)) | Potential hazards (see references in the text) |
|---|---|---|
| 1 | <ul><li>Seamounts 100-1000 m high and > 700 m below sea level (b.s.l.)</li><li>> 80% lavas and < 20% pyroclastic deposits</li></ul> | <ul><li>Lava flows</li><li>Obstacles for navigation (submarines)</li></ul> |
| 2 | <ul><li>Seamounts > 1000 m high and > 700 m b.s.l.</li><li>> 80% lavas and < 20% pyroclastic deposits</li><li>Developed shallow magma plumbing system (especially the larger ones), potentially leading to flank instability</li></ul> | <ul><li>Lava flows</li><li>Subaqueous eruption-fed density currents</li><li>Subaqueous eruption column</li><li>Pumice rafts</li><li>Large gas bubbles</li><li>Sector collapse</li><li>Tsunamis</li><li>Obstacles for navigation (submarines)</li></ul> |
| 3 | <ul><li>Seamounts < 700 m b.s.l.</li><li>> 60% pyroclastic deposits</li><li>+/- Developed shallow plumbing system (depending on seamount size)</li><li>Higher flank instability due to abundance of pyroclastic material making up the seamount</li></ul> | <ul><li>Lava flows</li><li>Subaqueous eruption-fed density currents</li><li>Subaerial PDCs</li><li>Subaqueous and subaerial eruption column</li><li>Pumice rafts</li><li>Sector collapse</li><li>Tsunamis</li><li>Obstacles for navigation (submarines)</li></ul> |
| 4 | <ul><li>Emerged seamounts (> 70 vol% submerged)</li><li>> 60% pyroclastic deposits</li><li>+/- Developed shallow plumbing system (depending on seamount size)</li><li>High flank instability due to abundance of pyroclastic material making up the seamount</li></ul> | <ul><li>Lava flows</li><li>Subaqueous eruption-fed density currents</li><li>Subaerial PDCs</li><li>Subaerial eruption column</li><li>Sector collapse</li><li>Tsunamis</li></ul> |
| 5 | <ul><li>Flat-topped seamounts (guyots)</li><li>Originally emerged seamounts drowned below sea level for erosion and subsidence, and cessation of volcanic activity</li></ul> | <ul><li>Obstacles for navigation (submarines)</li></ul> |

## 4 Results

### 4.1 Seamount morphology and growth stage

Seamounts in our study were classified based on their morphotype (simple cone, composite edifice, caldera, guyot; Fig. 2, Table 2) and growth stage (Stage 1 to stage 5; Table 3). Results for their abundance, distribution and exposure analyses are reported below (Figs. 3-6).

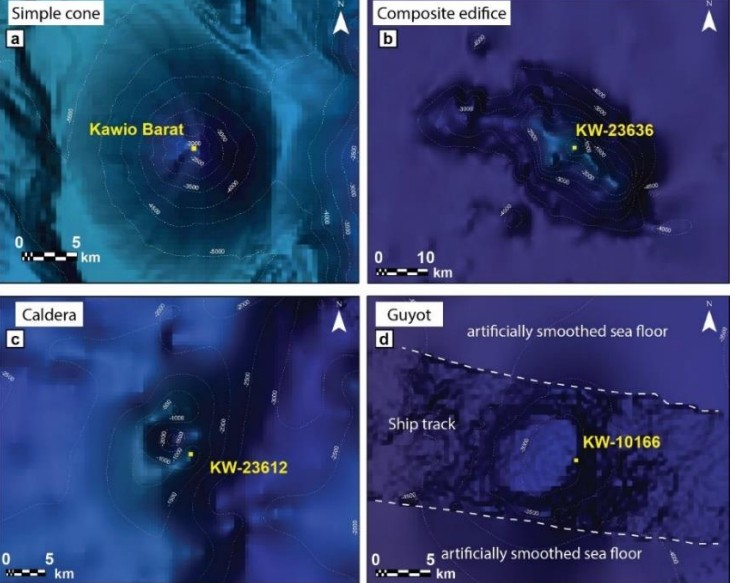

**Figure 2. Seamount morphotype examples. Note how the seafloor appears smoothed away from ship track measurements due to low density of data points (d). The basemap shown here is from NOAA (DEM Global Mosaic).**

Of the 466 seamounts in our catalogue, we were able to classify 295 (63%) of them into the four morphotypes; the remaining 171 (37%) seamounts were not classifiable due to low resolution bathymetry data and/or because they were too small. The seamounts are dominated by *composite* edifices (n= 255, 54.7%), followed by *simple cones* (n= 29, 6.2%), and guyots (n= 7, 1.5%). The morphotype least represented is calderas, with only 4 of them (0.9%). Water depths range from -5739 m b.s.l. (seamount KW-22106 of unknown morphotype; Lat: 13.81°, Lon: 125.58°) to 776 m a.s.l. (Paluweh, simple cone; Lat: -8.32°, Lon: 121.71°), with a *mode* of 2306 m b.s.l., *mean* -2106 m b.s.l., *median* -2221, and *skewness*= 0.02 (close to symmetric distribution) (Fig. 3c). Water depths within each morphotype category are relatively variable (Fig. 3d). Simple cones, composite edifices and calderas have a median of about 2000 m b.s.l., while guyots are mostly closer to sea level (median ~500 m b.s.l.), and unclassified seamounts cover the entire underwater range (~ 0-5700 m b.s.l.). Despite having a general broad range of water depths, all the classified seamounts are also represented at relatively shallow water depths (shallower than 1000 m b.s.l.), and this has important implications in terms of volcanic hazard (discussed in later sections). We found no particular geographic or tectonic setting distribution associated with each morphotype (Fig. 3a).

Results from the growth stage analysis (Fig. 3e) show that the majority of the seamounts in the study region are in their *stage 2* (65%, n= 303), > 1000 m high and > 700 m b.s.l., followed by the shorter but still deep *stage 1* (14%, n= 67), shallower *stage 3* (12%, n= 58), and emerged *stage 4* (7%, n= 31) seamounts. Only 7 seamounts (2%) represent *stage 5, flat-topped seamounts*. When comparing morphotype and growth stage distributions (Fig. 3f), simple cones and composite edifices are found in all growth stages, except for stage 5 (by definition), with composite edifices dominating across all stages. Calderas are only found within stage 2 (1%) and stage 3 (2%)

seamounts, however, when a caldera complex has new cones formed within them or on their rims, we classified them as composite edifices (e.g. Krakatau, Indonesia). Undefined seamounts dominate stage 1, however, they decrease in percentage towards higher stages seamounts. In terms of geographic/tectonic setting, stage 1 and 2 seamounts dominate extensional and/or intraplate domains such as back-arc basins (e.g. Banda Sea), and zones undergoing subduction (e.g. west of the Sumatra and Java trenches; east of the Philippines trench); while stage 3 and 4 seamounts are more common along volcanic arcs (e.g. Banda arc) and intraplate settings (Sunda Shelf, east of Vietnam; Fig. 1). An exception is represented by the South China Sea, an intraplate extensional setting, where the distribution of all grow stage seamounts is rather uniform, suggesting a more complex interplay of geological processes shaping seamount development.

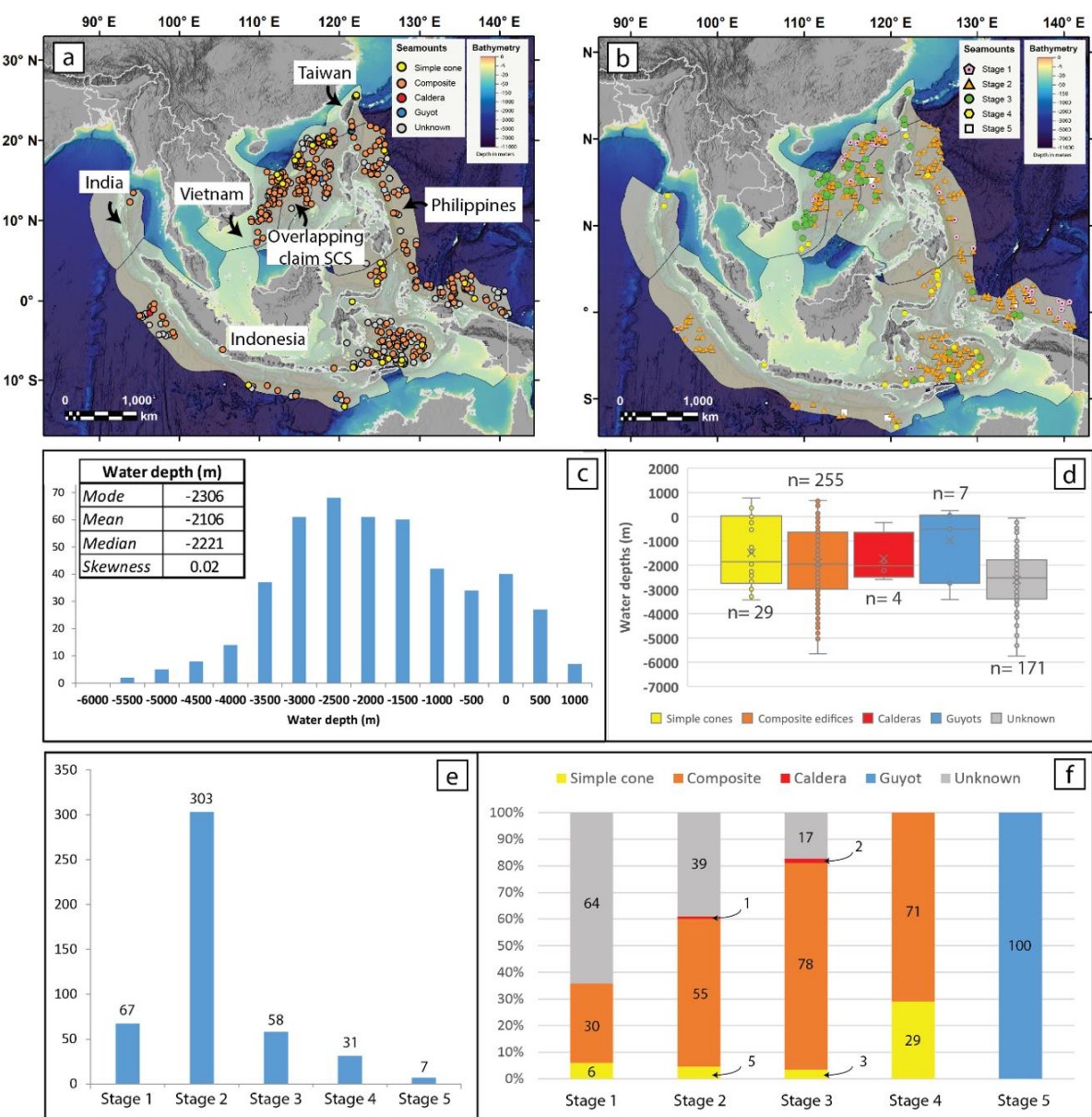

**Figure 3. Results of seamount classifications. Seamounts distribution maps based on their morphotype (a) and growth stage (b) (basemap is from NOAA - DEM Global Mosaic; yellow polygons represent exclusive economic zones). Distribution plots of water depths (c), and water depths vs morphotypes (d). Distribution of seamount growth stages (e), and normalised distribution (%) of seamount morphotypes within each growth stage (f).**

**4.2 Analysis of 'Exposure Potential'**

**4.2.1 Exposure Potential of assets around seamounts (quantitative)**

In this section we assess the exposure potential for population, submarine communication cables and ship traffic within 100 km radius from each seamount (Figures 4 and 5; Fig. S3 and Table S1). We found that 1.3% of the volcanoes of our catalogue (n= 6) have more than 1M people living within 100 km from them, with Huapinghsu (about 40 km north of Taiwan) exposing about 9.6M people, and two nearby volcanoes (Mienhuayu and Pengchiahsu) having a similarly high level of exposure (8.2M and 6.8M people), with Taipei lying approximately 60 km away. Krakatau volcano, Indonesia, also ranks high, with nearly 8M people exposed (Fig. 4, Fig. S3), many in Jakarta, which lies ~140 km to the east. About 8% (n= 39) of the seamounts expose between 100k and 1M people, and these are mostly located within the EEZs of Taiwan, Philippines and Indonesia. There are also a significant number of seamounts (n= 319) with zero population exposure, mostly located in the central portion of the northern South China Sea, western Pacific Ocean, and eastern Indian Ocean, and some in the Banda Sea.

Exposure for submarine cables has been evaluated in terms of total length of cables within 100 km from each seamount. About 50% (n= 232) of the seamounts have at least 50 km of submarine cables within their radii, and approximately 17% of seamounts (n= 78) expose more than 1000 km of cables each. The seamounts with higher exposure are within the EEZs of Taiwan, Philippines and Vietnam, with Taiwanese volcanoes exposing more than 2500 km of cables each (Fig. 4, Fig. S3).

Ship traffic density also shows the highest values around Taiwanese seamounts (Fig. 4, Fig. S3), with the busiest areas including the Taiwanese strait, western and eastern portions of the northern South China Sea (east of Vietnam and west of Philippines), Singapore and Malacca Straits, and Gulf of Thailand, with the last three having zero exposure due to lack of known seamounts nearby. Krakatau also ranks high for ship traffic exposure (11[th]).

In Figure 5, we aggregated exposure to the country level for individual seamount growth stages, and we found that 5 of the 11 EEZs in the region lie within 100 km of a volcanic seamount (India, Indonesia, Philippines, Taiwan and Vietnam), in addition to the central portion of the northern South China Sea, which is contended across different nations (i.e. here referred to as *overlapping claim waters*) and not discussed here. For population, Taiwan has the highest exposure values (up to nearly 10M people), followed by Indonesia (up to 8M) and the Philippines (< 1M). For exposure of submarine cables, Taiwan and the Philippines rank the highest (up to >2,000 km of cables nearby seamounts), followed by Vietnam, and the other EEZs having similar values, with overall less than ~1,500 km of cables within their maritime boarders. For ship traffic density, again, Taiwan reports the highest exposure, followed by the Philippines, Indonesia, Vietnam and India, with similar values respectively. When considering the growth stage, besides being the country with the highest exposure values, Taiwan is also the country with exposure to the seamounts with higher rank (stage 3 and stage 4), and this is shown for all the assets considered.

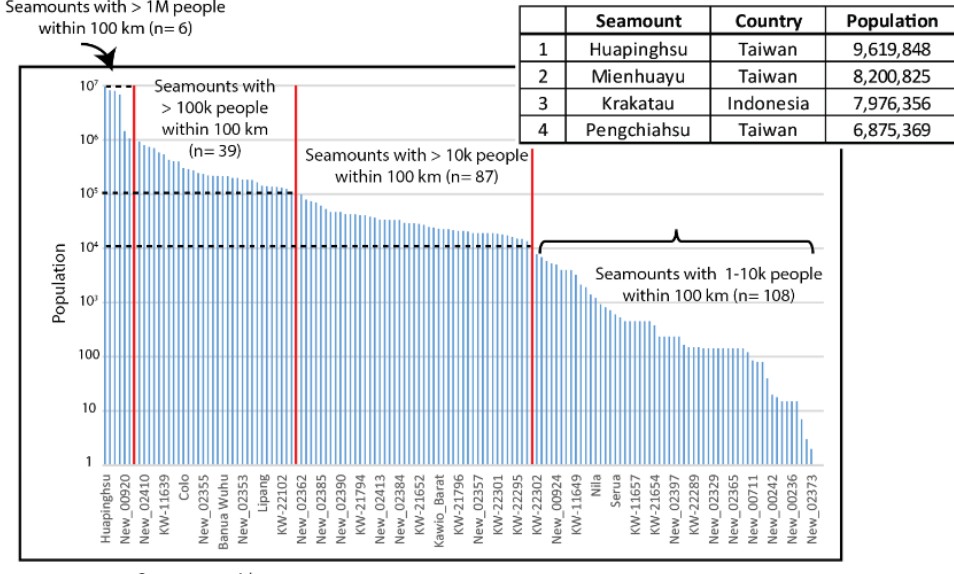

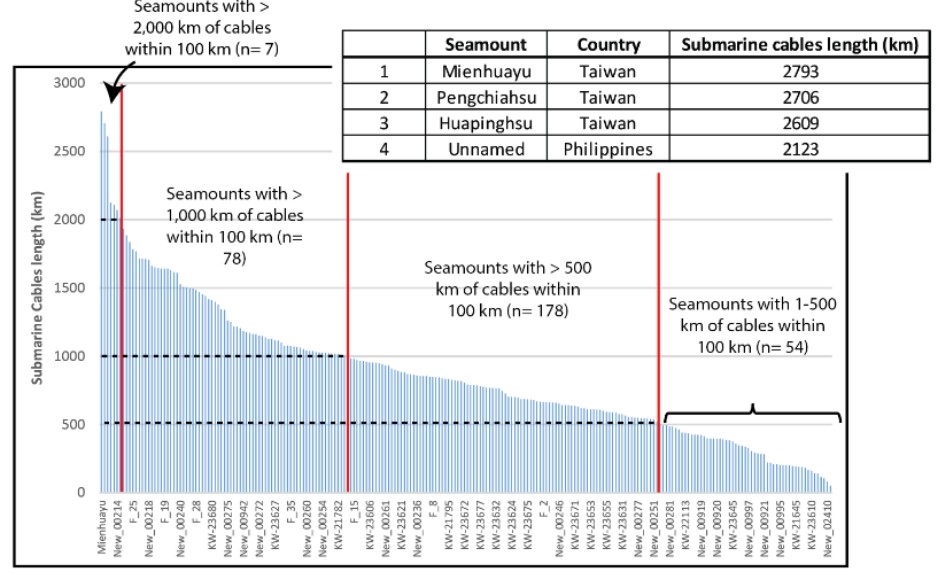

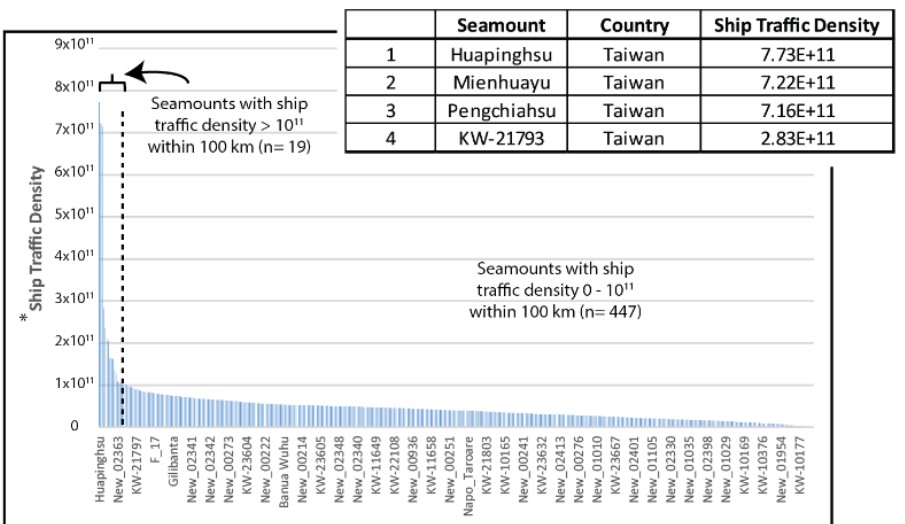

327

**Figure 4. Exposure potential for population (top panel), submarine communication cables (middle panel) and ship traffic density (bottom panel) within 100 km from seamounts in and around SEA. Tables with the top 4 seamounts for exposure are also reported (full seamounts exposure lists are available as additional material; Table S1).**

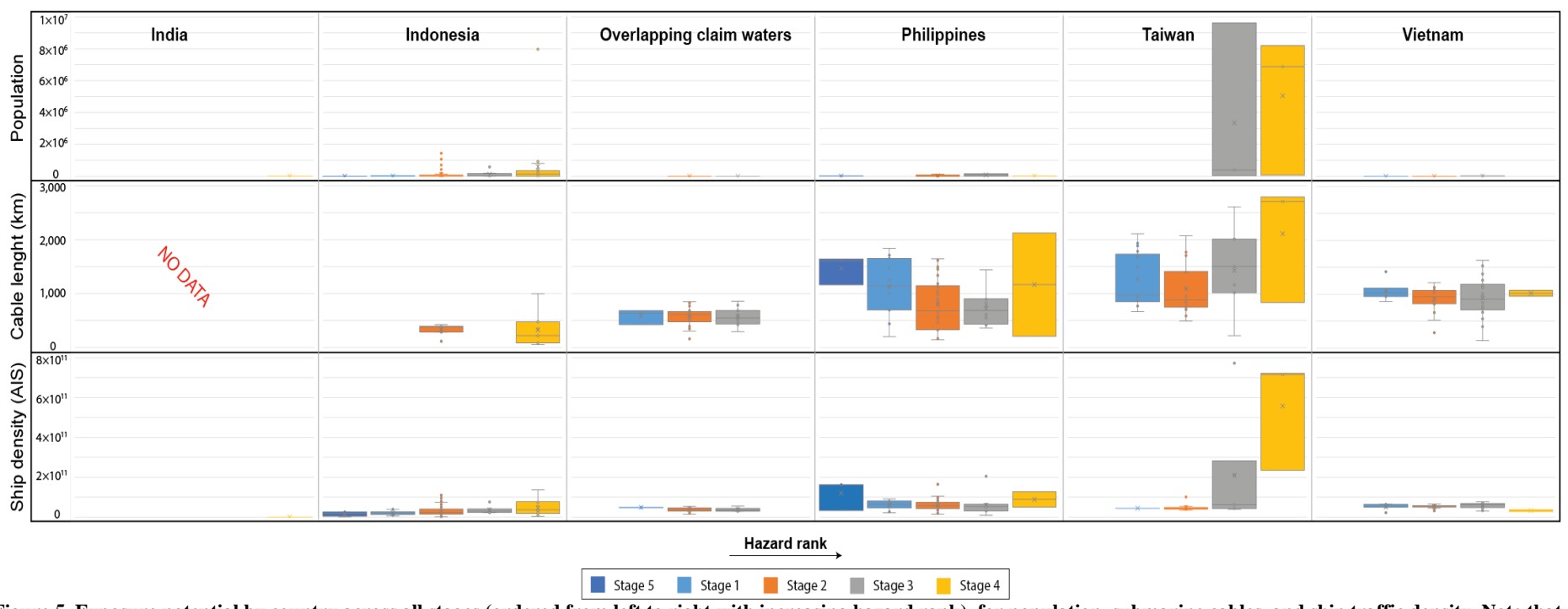

331
332 **Figure 5. Exposure potential by country across all stages (ordered from left to right with increasing hazard rank), for population, submarine cables, and ship traffic density. Note that**
333 **there is no cable data for India's seamounts**

**4.2.2. Hazard weighted seamount density (semi-quantitative)**

We conducted a weighted Kernel Density Estimation (KDE) to understand which regions have higher potential to produce hazards from a seamount. This estimation is purposely weighted towards the more hazardous seamounts (Stage 3 and 4) (more details about the weighting process are reported in the methods section). A sensitivity analysis was run with only stage 3 and 4 seamounts to test the effect of the lower-weighted stage 1, 2 and 5 seamounts, which represent the majority of the seamounts in our study, on the final weighted density map. Their effect was found to be minor (Fig. S4, Supplementary material), therefore we proceeded with this approach by including all seamount stages with a given hazard weight (Table 1). The results (Fig. 6) show that there are two large regions of interest, the largest is in the South China Sea, followed by the Banda Sea. Other areas of interest, but with lower density, include the Celebes Sea, the Halmahera Sea, and the portion of Pacific Ocean just east of Taiwan and northern Philippines. Countries surrounding the areas with higher weighted density include southern Vietnam, southern and northern Philippines, and eastern Indonesia (Sulawesi, Maluku, and Lesser Sunda Islands).

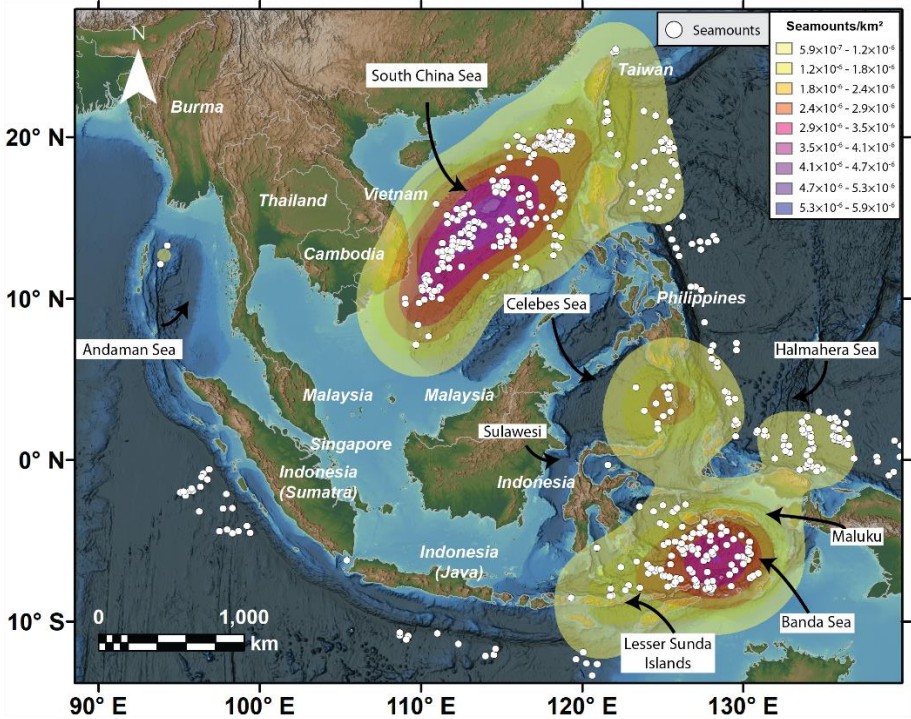

**Figure 6. Hazard-weighted seamount density map in the region of interest. In dark purple the area with higher density of seamounts of higher hazard (see Table 1 for hazard weight).**

**5 Discussion**

**5.1 Potential sources of volcanic and related hazards in Southeast Asia and its surroundings**

Our morphological assessment, while not incorporating geological, absolute age, and frequency/magnitude data (as further discussed in subsequent sections), offers valuable preliminary insights into the historical and possible future activities of seamounts. This approach is instrumental in identifying potentially hazardous seamounts for more detailed investigation in future research, as discussed below.

Mienhuayu and Pengchiahsu (offshore north of Taiwan), are two stage 4 simple cones, which lie in waters shallower than 200 m, with their summits just above sea level (16 and 49 m a.s.l. respectively). We can hypothesise that simple cones found in shallow waters in our region of interest are likely relatively young, because the sea

level rose about 120 m over the last ~20k years (Diekmann et al., 2008; Hanebuth et al., 2011). If volcanoes that are now in shallow water environments were already existing 20k ago, we could assume that they were at least partially above water, and this would be reflected in their shape (e.g. presence of prominent terraces on the flanks). Mienhuayu and Pengchiahsu, which have their base at ~200 m b.s.l., and are considered Pleistocene in Age (100 ka or younger: Global Volcanism Program 2013), would be good case studies to test this hypothesis, however, the current available resolution prevents us from providing reliable inferences at this stage. Focused bathymetric and/or seismic surveys around these volcanoes would provide key clues about their relative age (older or younger than the last glacial maximum, 25 ka). This is important for hazard assessment, because Mienhuayu and Pengchiahsu are 50-60 km of the Taiwanese mainland and are among the volcanoes that rank the highest in the quantitative exposure potential analysis for all the assets considered (Fig. 4) (more discussion in the following sections).

The Kawio Barat seamount (~100 km south of the Philippines; Lat: 4.68°, Long: 125.09°) is a large simple cone rising from about 5,500 m b.s.l. up to ~2000 m b.s.l. (stage 2); it is unlikely that such a high seamount was formed in a single or short-lived eruptive event. Its regular conical shape, its height and the relatively steep slope angles (up to >30°) suggest a past explosive or mixed explosive/effusive history, as observed at similar cones on land; therefore, it represents another candidate to attention for future studies.

In this study Krakatau has been classified as a composite and stage 4 seamount, even though it is the newest cone formed as part of a caldera complex. It is well known for the 2018 eruption collapse-tsunami event (Self, 1992), and for the catastrophic eruption of its predecessor in 1883, which produced PDCs and tsunamis, killing over 30,000 people (Self, 1992). An example of a less known but still potentially hazardous composite and stage 4 seamount in the region is North Kawio, Indonesia (northern portion of the Sangihe volcanic arc; Lat: 4.68°, Long: 125.47°). This seamount is reported as Pleistocene in the GVP, but no other information about age is provided. It is a mostly submerged edifice, with multiple peaks above sea level (e.g. Marore, Kawio, and Kamboreng islands) and several submarine vents, covering a total area of about 1,500 km$^2$. These characteristics (distributed volcanism), besides the unknown and possibly relatively young age, and the relatively close proximity to mainland southern Philippines (~100 km south), make North Kawio a potential seamount that could create tephra and tsunami hazards.

In terms of potentially more explosive submarine eruptions, calderas are key morphotypes to consider for the region. In our classification we identified 4 calderas, 3 of which have their summit at a water depth larger than ~1,300 m, with 2 of them being deeper than 2,000 m (all stage 2). Some calderas may form due to gradual subsidence over a longer period of time, hence not associated with any catastrophic explosive event. One key morphological indicator of either sudden or gradual collapse may be hidden in the intra-caldera slope angles; steep inner flanks may indicate a sudden sub-vertical movement downward resulting from the magma withdrawal from a shallow magma chamber. Despite the long-believed concept that explosive volcanic activity is hindered at large water depths (Cas, 1992), we show in our study that deep calderas with explosive features do exist. Seamount KW-23612, in the northern South China Sea (Fig. 2c), for example, despite having its rims reaching about 230 m b.s.l. (stage 3), has the caldera floor at over 2,000 m b.s.l., with inner slope angles up to ~50°. It is unlikely that such a depression (nearly 2,000 m deep), with such steep caldera walls was formed by gradual subsidence. Similarly, the recent eruption at the Hunga volcano, was responsible for deepening its caldera floor from an initial depth of about 200 m to about 850 m (Ribo et al., 2023). The eruption, although initiated at shallow water depth,

was responsible for the withdrawal of intra-caldera material up to >800 m through explosive mechanisms, and this has important implications, once again, about the water depth limit of volcanic explosive eruptions. It is clear that explosive activity associated with caldera formation can be of rather large magnitude, resulting in high hazardous scenarios, particularly if this occurs in highly populated areas such as the South China Sea (e.g. seamounts KW-23612, New-00258) or off the coasts of Indonesia (seamount KW-10401).

### 5.1.1 The geodynamic context for SEATANI seamounts

We noted that the geology, absolute age, and eruption frequency/magnitude, were not considered for our hazard-exposure potential assessment, because of the lack of information for most of the seamounts in the region. Notwithstanding, in this section we discuss the geodynamic context of the SEATANI seamounts from a regional perspective, with a particular focus on the two regions that we found to host the highest number of stage 3 and 4, hence potentially more hazardous, seamounts: the South China Sea and the Banda Sea.

A number of seamounts (Fig. 6) close to the Indonesian and Philippines trenches show zero to very low weighted density, and this reflects their relatively low hazard potential. Seamounts in these particular tectonic settings are likely millions of years old and no longer active, being at the end of their cycle and approaching a subduction zone (Staudigel and Clague, 2010). It is likely that also some seamounts in the high weighted density regions of the South China Sea and Banda Sea may have been extinct for millions of years, however, these areas are in tectonic settings, and must be discussed separately.

The South China Sea is the result of a multiphase continental rifting and breakup from the Eocene to the Miocene (e.g. Franke, 2013). Many studies provide evidence for extensive intraplate volcanism in the South China Sea following the end of the continental rifting (e.g. Xia et al., 2018; Gao et al., 2019; Zhao et al., 2020), with abundance of Late Cenozoic OIB-type basalts, inferred to be linked to a mantle plume (Yang et al., 2019). A more recent study identified widespread and partially still ongoing hydrothermal activity in the northern South China Sea, thought to be associated with magma intrusion (Zhao et al., 2021). On the southwestern edge of the South China Sea, east of southern Vietnam, there is a submarine volcano that erupted in historical times, Ile des Cendres, 1923. The volcano is located in proximity to a major regional fault, the East Vietnam Fault (Hall and Morley, 2004; Li et al., 2013) (Fig. 1); other major faults exist in the region, Mae Ping Fault and Three Pagogas Fault (Hall and Morley, 2004; Li et al., 2013), Fig. 1), and others may not be mapped, together with volcanoes in their proximity. Therefore, despite the intraplate setting, volcanism in the South China Sea may still play a role for future hazardous scenarios for the region.

The Banda Sea, on the other hand, results from more complex geodynamics. This area was formed by the initial collision between the Australian and the Banda arc (which was already active from ~12 Ma, (Yang et al., 2021), and subsequent slab rollback, which created the extensional Banda Sea back-arc basin (Wei et al., 2023). Therefore, seamounts in this area belong to at least two different formation mechanisms, arc volcanism and back-arc extensional volcanism. If we consider the seamount growth stage for this area (Fig. 3b), we notice that that majority of seamounts along the Banda arc are stage 3 and 4, while the seamounts in the central portion of the Banda Sea basin are stage 2. Most of these stage 2 seamounts are likely as old or older than 3 Ma, given the inference that volcanism in the Sunda back-arc basin ceased about 3 Ma (Honthaas et al., 1998), however, some of the Banda arc seamounts (i.e. Banda Api, Serua, Nila, and Teon) are reported in the GVP, and erupted in recent times (within the last ~120 years). Other seamounts are found along this arc and include a stage 3 (New-02400)

and a stage 4 (New-02393) seamount. Although we lack geological information from these two volcanoes, their
tectonic setting and proximity to other active seamounts may suggest that they were active in relatively recent
times and may still present a potential threat for the region in case of eruption.

**5.2 Benefit of multibeam high-resolution image analysis**
As part of the seamount characterisation, and where multibeam high-resolution (90-m/pixel) bathymetry data
were available, we searched for morphological features on and around seamounts that may give us clues about
past hazards (examples reported in Figure 7). Note that this information is reported here for discussion purposes
only, rather than for quantitative assessment of frequency and type of these past events because multibeam data
only covers < 10% of the region of interest, which may bias the results.
We identified several debris avalanche deposits, landslide scars and slumps, explosive craters at depth, new
potential seamounts, and deposits associated with submarine explosive volcanic activity. To understand the
potential of such past hazards, and the impact they would have if they occurred nowadays, we highlight an
example of a large landslide scar and associated deposit near Kawio Barat seamount, in the Celebes Sea, ~100
km south of Mindanao and 90 km north of Sangihe Island, Indonesia (Fig. 7b). We roughly estimated the debris
avalanche volume from the topographic contours (through Esri®ArcMap™ 10.7.1), which resulted in a volume
of  ~14 km$^3$ of material; the deposit includes visible blocks (i.e. hummocks) up to ~500 m in diameter, which are
typical of massive sector collapses (Violante et al., 2003; Idárraga-García and León, 2019; Carter et al., 2020).
For comparison, the sector collapse of Mt Krakatau volcano, Indonesia, in 2018, was about 0.15 km$^3$, which
produced a local tsunami with maximum run-up of up to 14 m, and caused over 430 fatalities and millions of USD
damage (Paris et al., 2020 and references therein). The event considered here is likely two orders of magnitude
larger than the Krakatau event, and although the associated potential tsunami hazard cannot be compared directly
because of bathymetric differences at both sites, the size of this event in the Sangihe arc gives us an idea of the
relative scale.
Slumps are generally considered less likely to produce significant tsunamis, however, in some instances they
have been inferred as the main cause of devastating tsunamis, such as the 1998 Papua New Guinea event (Okal
and Synolakis, 2003; Brune et al., 2010). Subaqueous slumps appear as transverse ridges with steep toes and block
of various sizes, as have been observed from bathymetric surveys around Hawaii (Moore et al., 1989), and in
contrast to debris avalanches, they are not associated with any amphitheatre-like detachment area. An example is
shown in Fig. 7a, where an area of over 100 km$^2$ of slumped material is highlighted, just north of seamount New-

470    00919.

Explosive craters provide evidence of volcanic hazards; in Figure 7c we report an example from seamount
KW-21797, ~300 km east-southeast of Taiwan and ~400 km northeast of Luzon, which is a composite and stage
2 seamount, and has a prominent topographic relief with a circular depression at the base of its NW side. This
topographic feature is at a water depth of about 4,600 m, has a crater diameter of approximately 1.5 km and is
around 150 m deep. We interpret this structure as a possible explosive crater because of its relatively large crater
diameter and rather regular circular shape, which may have been formed by an individual explosive event. To the
west of this structure, we identified an apron-like morphology extending westward for about 4 km, which is likely
the volcanic deposit associated with this explosive structure mantling its flank. Though, we cannot rule out the
possibility that this structure and associated deposit may be related to effusive activity forming a westward lava
flow. Evidence of explosive volcanism at water depths ≥ 1000 m has been reported in literature, both along
volcanic arcs (Murch et al., 2019b), mid-ocean ridges (Sohn et al., 2008), and hotspots (Schipper et al., 2010).
Additionally, the potential occurrence of explosive deep-sea volcanic eruptions has been proved through analogue
experiments (Dürig et al., 2020; Newland et al., 2022; Head and Wilson, 2003).
In the same area of seamount KW-21797, we identified other possible seamounts (Fig. 7d) that are not reported
in any official dataset. They are individual composite edifices or chains of composite edifices (at least 3 chains
can be recognised, all extending along W-E trends). These potential seamounts vary in height from < 500 m to
~1500 m, and their summit reaches water depths of ~5,500 to ~4,000 m b.s.l. Although all these seamounts have
their summit in deep waters, some of them are higher than 1,000 m (stage 2). We do not include them in SEATANI
as we cannot be sure that they are volcanic, but they may be worth further investigation.

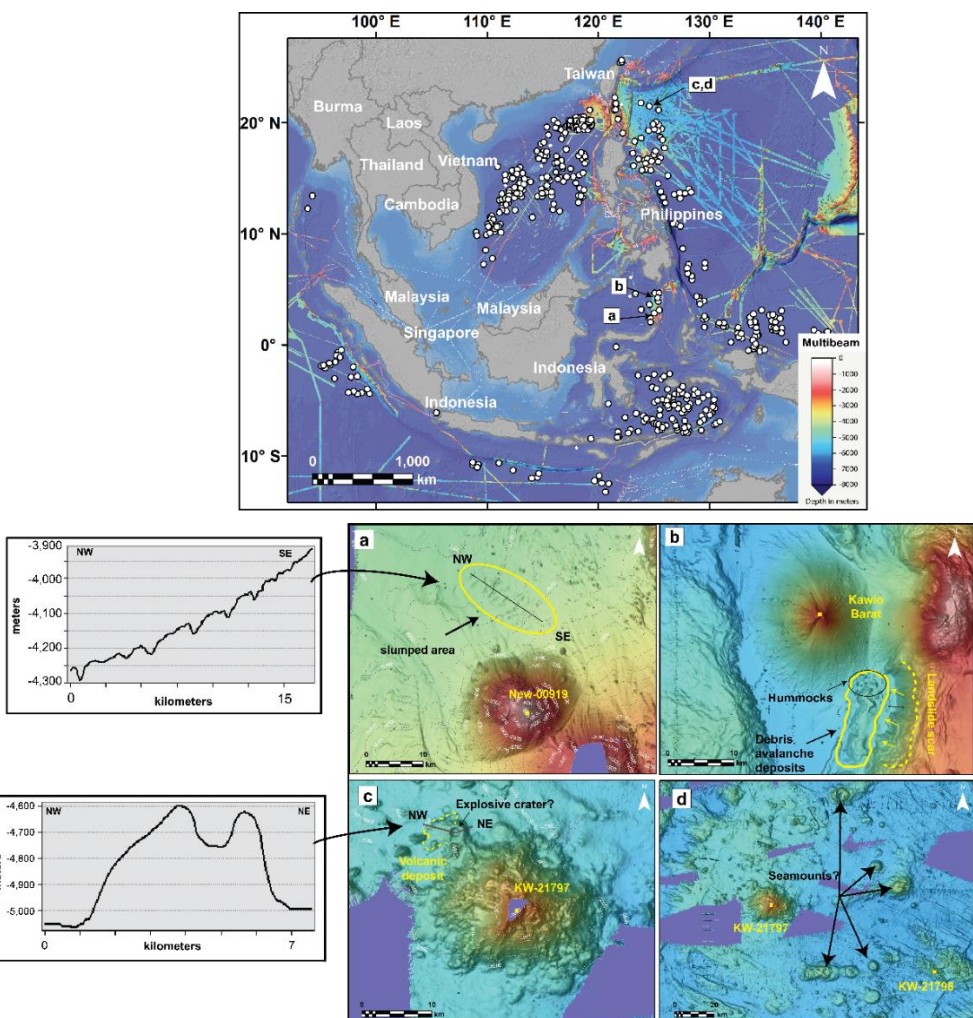

**Figure 7. Map of the study region with multibeam data coverage (top panel), and relevant hazard features**
**at some of the seamounts investigated, where multibeam data were available (bottom panels, a to d). Two**
**bathymetric profiles for box a) and c) are also shown.**

## 5.3 Countries with high hazard/exposure potential

Our seamount characterisation and exposure potential analyses highlights areas potentially more exposed to
hazards in case of submarine eruption in and around SEA. Taiwan seems to be the candidate that requires more
attention. It ranks high for both exposure analysis types conducted here. It has the highest number of people
exposed, with two stage 4 and one stage 3 seamount (the most hazardous types) just 30-60 km northeast of the
highly populated Taipei District (>9 M people). A submarine eruption at such distances may affect the nearby
Taiwan through tephra falls and tsunamis. Subaqueous landslides, PDCs, or lava flows can damage the dense
submarine cables array both north and south of Taiwan. Both volcanic and tsunami hazards can affect local ship
traffic, which seems to be the densest in the whole region, with key connection between Taiwan and the rest of
the region through eight major ports.
Besides Taiwan, if we consider the exposure by number and type of seamount by country (Fig. 5), Indonesia,
Philippines and Vietnam are potentially threatened too. For Indonesia, the well-known Krakatau volcano is a
hazard for population (~8 M people) and ship traffic, being a key passage to the South China Sea from the southern
Indian Ocean. Eastern Indonesia (Sulawesi, Maluku and Lesser Sunda Island) is mostly exposed to stage 4, stage
3, and stage 2 seamounts. The Philippines is highly exposed as well, with the maximum exposures in the north
(submarine cables, coasts), west (ship traffic) and south (coasts). Vietnam is characterised by high ship traffic
density, with major commercial areas including the Mekong delta and Da Nang port. The Vietnam EEZ encloses
a seamount that erupted in historical time, Ile des Cendres, 1923 (~115 km off the southeast coasts of Vietnam;
Lat: 10.16°, Lon: 109.01). The eruption formed two islands (eroded soon after the eruption and now completely
underwater) and at least another submarine cone (Global Volcanism Program, 2013). There is not much
information about the eruption, but it was thought to be VEI= 2, and a local tsunami along the SE coasts of
mainland Vietnam was also reported (Vu, 2008; Dai Dien, 2010). To our knowledge, there is no detailed study of
the Ile Des Cendres complex, despite it representing the latest episode of submarine volcanism in the region and
having a distributed nature (formation of several vents), which increases the area from which a potential eruption
may occur, hence the hazard.
In areas of low seamount density and apparent lower hazard - for example the Indian EEZ (Andaman Sea,
between Sumatra and Burma), which contains just two seamounts, both stage 4 - tsunami can potentially affect
wider areas, such as the coasts of Thailand, Malaysia, Burma, Indonesia (Sumatra), India and, depending on the
magnitude of the event, the coast of Singapore through the Malacca Strait. One of these seamounts is Barren
Island, which shows past evidence of sector collapse (Chandrasekharam et al., 2009). Given the tectonic setting
of these two volcanoes (submarine continuation of the Indonesian volcanic arc), we may expect the presence of
other seamounts in this area not currently charted, hence potential increased hazard extent.

**5.4 Limitations and future goals**
A major limitation of this study is the fact that we characterise the hazard potential from seamounts solely based
on morphological (morphotype) and structural information (i.e. water depths, heights), with the high likelihood
of including volcanoes that might have been inactive for millions of years, in turn resulting in an overestimation
of the hazard potential. Despite this, the present study is relevant because it provides the elements to narrow down
the research for future hazard studies from submarine volcanic activity in and around SEA.
The choice of the main seamount dataset used for our analysis, that from Gevorgian et al. (2023), was justified
by at least two reasons: (1) it is the most up-to-date seamount dataset in terms of data quality of the Vertical
Gravity Gradient (data noise reduction of 40% from previous VGG datasets; Gevorgian et al., 2023); and (2) it
only focuses on volcanic seamounts, not including small volcanic features such as knolls, which are negligible in
terms of volcanic hazards, and other non-volcanic large features. Even though their method excludes potential
seamounts from within continental margins, potentially biasing our results toward lower hazard potential from
areas such as the Sunda Shelf, we consider it a first approach to be improved upon with higher resolution data.
The use of other global seamount datasets (Cañón-Tapia, 2023, and references therein), which include a higher
number of seamounts (e.g. 33,452 seamounts and 138,412 knolls; Yesson et al., 2011), may have the opposite
effect leading to an overestimation of the hazard potential. Future studies should work to integrate the multiple
other seamount datasets, supported by higher resolution bathymetry datasets, when made available, which will
also help to characterise the large number of unclassifiable seamounts (n= 171) from this study.
On a parallel note, here we focused on submarine volcanism and excluded large volcanic islands and coastal
volcanoes, however, we acknowledge that similar hazards to those produced by seamounts can be generated by
such volcanoes, and extensive work has been already carried out in SEA (Zorn et al., 2022, and references therein).
Therefore, future work can take advantage of both approaches for a more comprehensive assessment for the
region.
The lack of seamounts identified on the Sunda shelf using our method and datasets is a shortcoming of using
the Gevorgian et al. (2023) dataset that filters out seamounts near continental margins. The known submarine
volcanoes of Ile des Cendres and Veteran, alongside terrestrial volcanoes not reported in this study (e.g. Ly Son
group, Table S2), are in close proximity with a major regional fault, the East Vietnam Fault. Since this fault
extends across the central-eastern portion of the Sunda shelf (all the way south to Borneo inland) (Li et al., 2013),
it would not be surprising to have other volcanoes along or near to this fault zone. Other major faults mapped on
the Sunda shelf include the Wang Chao (Hall and Morley, 2004) and Three Pagogas faults (Li et al., 2013), but
no seamount is known to exist around these areas. The number and type of seamounts potentially not mapped and
not considered for this study, may bias our results, particularly with regards to the KDE assessment. Potentially,
the threat to countries not currently considered exposed, e.g. Singapore, is much greater than currently
appreciated, because of the lack of continental shelf mapped seamounts. However, once again, we emphasize that
here we did not produce any volcanic hazard maps for the region, but rather conducted a preliminary assessment
of hazard and exposure potential, highlighting seamounts and areas of interest that can be the focus for more-in-
depth studies.
When it comes to explosive versus effusive behaviour of a given volcano, hence the type of hazards it can
produce, magma composition is a key aspect to consider. In subaerial environments, more silicic magmas are
generally more explosive than basaltic magmas. However, in subaqueous environments the interaction between
external water and magma is often consider the leading trigger of the explosivity of that particular volcano (e.g.
Verolino et al., 2018, 2019). Many pioneer studies on the topic showed that this explosive interaction is more
likely to occur with basaltic magmas (e.g. Wohletz, 1983, 1986; Büttner and Zimanowski, 1998), nevertheless, it
also occurs with more silicic compositions (e.g. Austin-Erickson et al., 2008; Dürig et al., 2020). Magma
composition was not accounted for in our assessment of hazard-exposure potential for two main reasons: 1) Only
GVP seamounts have known composition (despite it could be assumed for some of the seamounts based on their
tectonic setting); And 2) explosivity in subaqueous settings have been observed/inferred across all compositional
domains, hence producing similar hazards regardless of composition. However, one difference is the production
of pumice rafts in silicic eruptions (e.g. Havre, 2012; Fukutoku-Oka-no-Ba, 2021; Carey et al., 2014; Maeno et
al., 2022), which is not expected for basaltic eruptions. Magma composition, eruption dynamics, and
environmental factors that affect hazard extent, distribution and intensity, such as wind conditions or bathymetry,
should be accounted for in future quantitative hazard studies for the region once more information is made
available.
Two main issues about the study of seamounts globally and regionally are that 1) the detection from space is
limited within continental margins, and 2) the currently available bathymetry resolution is not enough to allow a
comprehensive morphological characterisation of seamounts. As a result, we end up with large areas without
seamounts (e.g. Sunda shelf), and many unclassified seamounts (n= 171).
For the quantitative exposure analysis, we used a 100 km radius around each seamount to indicate areas that
may be impacted by volcanic activity. However, concentric radii, despite used in previous hazard studies, are not
a necessarily good approximation of how volcanic hazards behave (Jenkins et al., 2022): some hazards may affect
areas smaller (e.g. lava flows, PDCs) or larger than the 100 km radius (e.g. tephra fall, pumice rafts).
Another limitation regards the exact location of submarine communication cables and how many people rely
on this technology. The communication companies provide station to station information, which means that the
exact path of each cable may not be as reported, and this probably partially affects our exposure results.
Additionally, all countries in our study region depend on submarine cables for internet use, which translates into
over 600 million people in the region, however, the cable length analysed here does not give a direct information
on the potential impact from a submarine volcanic eruption, which would be provided, for example, by the exact
number of people that rely on specific cables per country. Despite this limitation, the direct relationship of
seamount and cable density in some areas (northern South China Sea, Luzon Strait, East China Sea) is rather
obvious (Fig. S2) and should be accounted for with regards to future cable installations in the region.

The above limitations can be overcome in different ways. One is to improve our collaborative effort with private
and government agencies, which may have seismic and bathymetry data that may improve our understanding of
volcanic hazards from submarine volcanoes in the region. Another is to improve the existing bathymetry datasets,
by combining direct bathymetric information from Gebco and from local nautical charts (Felix et al., 2022); this
will help with a better regional seamount characterisation, hazard assessment, and eventually hazard modelling
and impact analysis at key locations. A third possibility is to use new satellite altimetry data of the sea surface,
which will be made available from NASA in 2024 through the SWOT (Surface Water and Ocean Topography)
mission, which was launched in December 2022. These new data will provide unprecedented resolutions of the
sea surface, which in turn will be used to estimate location of smaller seamount than those currently detectable
from satellite-derived methods, at a global scale. These data could be combined with the bathymetry data for more
comprehensive analyses of hazard. Lastly, the results reported in this work, in addition to new data, will provide
an evidence base for more focused investigations to be conducted at potentially high threatening seamounts
(including sampling through Remotely Operated Vehicles, and later laboratory analysis for a complete
characterisation). This will serve countries across the region to become more prepared and resilient against
submarine volcanic hazards.
**6 Conclusions**
Seamounts are an understudied and potentially silent and unseen threat for human populations and infrastructure.
Despite the global identification of about 35,000 seamounts (Gevorgian et al., 2023), only a few of them are
thoroughly studied and monitored (e.g. Deardorff et al., 2011; Caress et al., 2012; Carey et al., 2014; Berthod et
al., 2021). We conducted a seamount characterisation and associated hazard-exposure potential assessment on a
regional scale for SEA and surrounding areas, through the SEATANI dataset, which provides the basis for more
focused investigations of hazards for the region in the future at key locations. Our results show that composite
and stage 2 seamounts are the most abundant in the region, however, stage 3 and stage 4 seamounts (simple,
composite and calderas) are the most important for hazard potential and numbers of people, lengths of cable and
density of shipping exposed. Taiwan has the highest total exposure potential (across all exposure types) within
100 km of volcanic seamounts, followed by Indonesia, Philippines and Vietnam. The hazard-weighted seamount
density assessment highlights two main areas of interest: the northern South China Sea and the Banda Sea. Any
volcanic and related hazards (e.g. tsunamis), if generated in these areas, will potentially affect the coasts of
Southern Taiwan, northern and southern Philippines, Vietnam and eastern Indonesia.
This work represents the first step towards understanding the threat that submarine volcanoes pose to
populations and infrastructure in and around SEA. The integration of new bathymetry, seismic and satellite-
derived altimetry data (i.e. SWOT mission) will shed more light on the potential of these volcanoes and enhance
awareness, preparedness and resilience for the countries surrounding these waters.

**Data availability**

Data are available in the supplementary material files and in the public data repository of NTU
(https://researchdata.ntu.edu.sg/privateurl.xhtml?token=820ea7c9-4ff4-48f8-8e8b-98cd4ffe01f8)

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

## Acknowledgements

We would like to thank the editor G. Macedonio and the reviewers E. Cañón-Tapia and E. Nicotra for improving
this manuscript. This research was supported by the Earth Observatory of Singapore via its funding from the
National Research Foundation Singapore and the Singapore Ministry of Education under the Research Centers of
Excellence initiative. This work comprises EOS contribution number 531.

## Author contribution

AV: Paper conceptualisation and preparation, figures production, data elaboration, analysis,
and interpretation, editing; SFW: data elaboration, analysis, and interpretation; SJ: paper conceptualisation,
editing; FC: paper conceptualisation, editing; ADS: paper conceptualisation, editing.

## Competing Interests

We declare no competing interests.