# Peer review of "SEATANI: hazards from seamounts in SouthEast Asia, Taiwan,"

_EGUsphere, 2023_

## Referee Comment (RC2)

**SEATANI: hazards from seamounts in SouthEast Asia, Taiwan, and Andaman and Nicobar Islands (eastern India)**

Andrea Verolino ( ✉ andrea_verolino@hotmail.it )
  Earth Observatory of Singapore   https://orcid.org/0000-0002-9335-3993

**Su Fen Wee**
  Nanyang Technological University

**Susanna Jenkins**
  Nanyang Technological University   https://orcid.org/0000-0002-7523-1423

**Fidel Costa**
  Institut de Physique du Globe de Paris

**Adam Switzer**
  Nanyang Technological University   https://orcid.org/0000-0002-4352-7852
* * *
**Article**

**Keywords:**

**Posted Date:** September 18th, 2023

**DOI:** https://doi.org/10.21203/rs.3.rs-2950249/v2

**Additional Declarations: Yes** there is potential Competing Interest. In order to avoid any conflict of interest, we declare that co-author A.D. Switzer is a member of the editorial board for Communications Earth & Environment (Ocean and Cryosphere).
* * *
**SEATANI: hazards from seamounts in SouthEast Asia, Taiwan, and Andaman and Nicobar Islands (eastern India)**

[revised manuscript text omitted]

| Seamount growth stage | Description (from Staudigel and Clague, 2010)) | Potential hazards (see references in the text) |
|---|---|---|
| 1 | ● Seamounts 100-1000 m high and > 700 m b.s.l.
● > 80% lavas and < 20% pyroclastic deposits | ● Lava flows
● Obstacles for navigation (submarines) |
| 2 | ● Seamounts > 1000 m high and > 700 m b.s.l.
● > 80% lavas and < 20% pyroclastic deposits
● Developed shallow magma plumbing system (especially the larger ones), potentially leading to flank instability | ● Lava flows
● Subaqueous eruption-fed density currents
● Subaqueous eruption column
● Pumice rafts
● Large gas bubbles
● Sector collapse
● Tsunamis
● Obstacles for navigation (submarines) |
| 3 | ● Seamounts < 700 m b.s.l.
● > 60% pyroclastic deposits
● +/- Developed shallow plumbing system (depending on seamount size)
● Higher flank instability due to abundance of pyroclastic material making up the seamount | ● Lava flows
● Subaqueous eruption-fed density currents
● Subaerial PDCs
● Subaqueous and subaerial eruption column
● Pumice rafts
● Sector collapse
● Tsunamis
● Obstacles for navigation (submarines) |
| 4 | ● Emerged seamounts (> 70 vol% submerged)
● > 60% pyroclastic deposits
● +/- Developed shallow plumbing system (depending on seamount size)
● High flank instability due to abundance of pyroclastic material making up the seamount | ● Lava flows
● Subaqueous eruption-fed density currents
● Subaerial PDCs
● Subaerial eruption column
● Sector collapse
● Tsunamis |
| 5 | ● Flat-topped seamounts (guyots)

[revised manuscript text omitted]
, although we did not consider geological, absolute age, and frequency/magnitude information in our seamount assessment (see discussion in later sections), can still provide insights about past and potentially future seamount behaviour. This can be used to narrow down the search of possible hazardous seamounts to investigate further in future studies 
[revised manuscript text omitted]

Two main issues about the study of seamounts globally and regionally are that 1) the detection from space is limited within continental margins, and 2) the currently available bathymetry resolution is not enough to allow a comprehensive morphological characterisation of seamounts. As a result, we end up with large areas without seamounts (e.g. Sunda shelf), and many unclassified seamounts (n= 171). The general lack of seamounts on the Sunda shelf is questionable. Ile des Cendres and Veteran, besides other volcanoes not reported in this study because not in line with the definition of seamount used here (
[revised manuscript text omitted]

**Supplementary Files**

This is a list of supplementary files associated with this preprint. Click to download.

- Supplementaryinformation.docx
- TableS1.xlsx
- TableS2.xlsx

---

## Author Response (AR1)

**Letter to editor**

Dear Dr. Giovanni Macedonio,

We are pleased to submit the revised version of "SEATANI: hazards from seamounts in SouthEast Asia, Taiwan, and Andaman and Nicobar Islands (eastern India)".

We have applied most suggestions proposed by the reviewers and/or added relevant additional text.

Our responses are reported below in red, with line numbers reported where changes were applied.

P.S. Please use All Markup > Balloons > Show All Revisions In Line, to have the same line numbers.

Best regards,
Dr Andrea Verolino et al.

**Reviewer 1: Edgardo Cañón-Tapia**

Database used for the analysis.

The base of the hazard estimation reported in the current submission is the dataset of Gevorgian et al. (2023), which is an updated version of the dataset of Kim and Wessel (2011). As noted recently by Cañón-Tapia (2023, Geoscience Frontiers), there are other databases that can be used for analysis like this one. In particular, the database of Yesson et al (2011) would have been useful in the present context to bracket hazard estimates. When the knolls and seamounts databases of Yesson are examined, there are 13696 features within the area of the current study, which is a much larger number than the 466 seamounts analyzed in the current submission. The results of the hazard analysis based on the location of such a larger number of possible seamounts would therefore be very different from the results reported in the work by Verolino et al.

As elements for the discussion on this subject, it can be noted that in the GeoscFront2023 paper, it was shown that both the Kim-Wessel, (and consequently the extension by Gevorgian, hereafter GKW) and Yesson et al (hereafter Yetal) datasets are based on the same gravity signal. The difference in number of the reported features is related with the filters applied to that signal. Perhaps the most equilibrated assessment of both datasets is something like this: GKW over-filtered the signal, leaving only a very small proportion of seamounts, whereas Yetal under-filtered the signal allowing the introduction of some noise. Within that context, it is extremely important to note that one of the imposed filters in GKW was precisely to eliminate signals in continental margins and their vicinity. Therefore, that database has BY DESIGN, only a small fraction of seamounts in the region studied by Vitalino et al. This aspect of the dataset used for the current analysis presented by Vitalino et al. therefore leads to an underestimation of the hazard potential in the area of study. It can be argued that the Yetal dataset would lead to an overestimation of that hazard potential because it includes many places that might not be true volcanoes. Ideally, the bracketing of hazard estimation using the Yetal dataset should be included in the revised version of the work by Vitalino et al., but if this is not practical for logistic/financial support reasons, I think that at the very least the work by Vitalino et al should mention the possibility of finding a different hazard estimate using a different database, leaving the door open for a future study that is made using the Yetal dataset and that offers an upper bracket of current hazard estimation in this region.

This has been addressed in Section 5.4 "Limitations and future goals" (lines 584-595: "*The choice of the main seamount……from this study."),* where we discuss the rationale behind our choice for the

Gevorgian et al. (2023) seamounts dataset, the difference that the datasets would be expected to produce for our results, and also acknowledge the utility of other datasets for future studies.

Method of analysis

2a)  KDE implementation

Although the current version of the paper by Vitalinom et al. states that details of the weighting process are given in the methods section, there is no factual information about that process that can be consulted by the reader, other than a statement indicating that KDE was performed on ESRI ArcMap 10.7.1 that uses a default bandwidth (no mention of how the weighting was managed within the calculations made by KDE). Although such ommision can be corrected very easily, a main subject of discussion should be the reliability of the default bandwidth on the ArcMap software. As discussed at length by Cañón-Tapia (2020 Geomorphology, 2021 ESR and especially 2022 FEART and the November 2023 issue of the BSGM), method selection may introduce unsuspected biases on the results of an analysis of spatial distribution. In particular, the automatized selection of a bandwidth parameter may not be adequate to gain a complete picture of the characteristics of the distribution. In the current case under discussion, the bandwidth selected by the software clearly includes an influence of the large areas in which there are not observations. Some of those areas are the result of a too restrictive database (as mentioned above), but other areas without data are the natural result of the presence of emerged lands. Thus, ArcMap calculates a blanket bandwidth that is too large for the small population of volcanoes, therefore biasing the results towards a too large scale distribution. As a result, the hazards associated with the groups of volcanoes on the southwest (actually all along the east Java and Sumatra coasts) are entirely sub-estimated, whereas those on the north and southeast are somewhat overestimated. Also, it must be noted that the automatically selected bandwidth (>600 km) is in contrast with the interest mentioned by the authors on the 100 km limit. This discrepancy on the scale of different stages of the analysis should be discussed in a revised version of the current submission.

To address these issues, it is recommended that the results of an exploration of the KDE are reported, in which the bandwidth is manually selected within the 50 to 600 km (increments of 50 km should suffice), to identify the scales at which hazards (concentrations of volcanoes and proximity with human-made infrastructure) may be changing in different sections of the area of study. Given the small number of volcanoes used in the currently used dataset, the general trends may not change much from those already reported, except from the possibility of increasing hazard level in the SW. Anyway, as mentioned above, this would be a lower bracket of hazard estimation, and this aspect of the analysis should be clearly highlighted in the revised version. The results of a similar analysis completed with a much larger population of seamounts is likely to produce entirely different results, which correspond to an upper bracket on the hazard estimation.

We explored other bandwidths for the KDE by conducting sensitivity analyses. In particular, a statement has been added to the manuscript in the method section 2.4 "Exposure Analysis" (lines 216-219: "To verify.….default bandwidth"), and the results with associated discussions have been included in the supplementary information file under "Choice of bandwidth for the KDE" (Fig. S2 and lines 186-199: "The choice of a suitable…..most appropriate choice for this study."). A highlight from this analysis is that the default bandwidth provided in ArcMap was suitable for the purpose of our work, which is a regional assessment of hazard potential, while we show that the choice of a small bandwidth (100 km) will be more appropriate for local studies at the scale of individual volcanic fields.

2b) Elimination of large emerged volcanoes.

Although this criteria eliminated only 16 volcanoes, it is strange that those volcanoes were eliminated on a work focusing on hazard estimation. Large eruptions from emerged volcanoes can lead to the entrance of pyroclastic deposits to the sea. Recent examples from Montserrat come to mind. Also, the potential for tsunamis generated by either pyroclastic deposits entering shallow waters and by (perhaps less likely) sector collapses of large edifices should be considered. The historic eruption of Krakatoa and some documented tsunamis on the islands of Hawaii come to mind. Indeed, this type of danger is different from that forming the bulk of the reported work, but I think it deserves at least some mention somewhere in the text.

Our threshold of 30% emerged edifice and 1000 m a.s.l. was somewhat arbitrary, but was targeted to maintain the focus on the bulk submarine portion of the edifice, while allowing reproducibility for future studies. We have now added a statement in the method section 2.1 "Compilation of SEATANI" to clarify this aspect (lines 132-133: "*Therefore, in order to guarantee reproducibility and to maintain our broad focus on the unknown hazard potential of seamounts, we …*"). Additionally, we acknowledge that coastal/island emerged volcanoes can cause similar hazards to submarine volcanism, and we included a note in section 5.4 "Limitations and future goals" (lines: 596-600: "*On a parallel note, here we focused on submarine volcanism…. for a more comprehensive assessment for the region.*")

**Reviewer 2: Dr Eugenio Nicotra**

- Within the volcanic seamounts classification, which is a qualitative analysis of the shape of the seamount, I think that the importance of the ridge/fissural/linear shape is a little bit underestimated. South China Sea and Banda Sea are two geodynamic settings characterized by continental rifting and back-arc extension, two geodynamic settings which preferably generate volcanic ridge. A typical example has been recently studied in the Tyrrhenian Sea (back-arc setting), with the new bathymetric model for Marsili seamount (Nicotra et al., 2023). So, if data available to the authors are defined enough, it could be useful also to evaluate the elongation of the seamounts.

In this work we did not make a distinction between the different types of composite edifices, because we thought it too ambitious for the level of bathymetry resolutions available. However, the different types of composite edifices were previously mentioned in the manuscript (Table 2 and Supplementary material file, lines 37-38: "*This category includes irregularly shaped multi-vent edifices, volcanic ridges, and cones formed on flanks of larger edifices.*"). We have added relevant text to the supplementary material file in the "Composite edifices" section, with relevant examples from outside and inside our study region (suppl. mat. file; lines 55-65: "*An example of a high hazard…..targeted hazard mitigation strategies*").

- On my opinion, section 5.3.1 about the geodynamic context is not useful at that point of the manuscript. So: 1) or it is moved in a background section after Introduction; 2) or it becomes a point of discussion in and background in the 5.1 section

Thank you for this comment. We moved 5.3.1 to 5.1.1, maintaining it as discussion rather than background information, because it is based on our results.

- Authors well know the limitation of their work (and also dedicate the 5.4 section to this), mainly due to the association of geological objects having millions of years of difference. Although results are very interesting, maybe the introduction is a bit over overloaded about the importance of this paper in terms of hazard from seamounts. A lot of work is still needed, but this culd represent a first step in the costruction of a useful hazard database for seamounts of the South-East sea.

We slightly shortened the introduction, where suggested by the reviewer.